# Nuclear export receptor CRM1 recognizes diverse conformations in nuclear export signals

**Ho Yee Joyce Fung, Szu-Chin Fu, Yuh Min Chook***

Department of Pharmacology, University of Texas Southwestern Medical Center, Dallas, United States

**Abstract** Nuclear export receptor CRM1 binds highly variable nuclear export signals (NESs) in hundreds of different cargoes. Previously we have shown that CRM1 binds NESs in both polypeptide orientations (Fung et al., 2015). Here, we show crystal structures of CRM1 bound to eight additional NESs which reveal diverse conformations that range from loop-like to all-helix, which occupy different extents of the invariant NES-binding groove. Analysis of all NES structures show 5-6 distinct backbone conformations where the only conserved secondary structural element is one turn of helix that binds the central portion of the CRM1 groove. All NESs also participate in main chain hydrogen bonding with human CRM1 Lys568 side chain, which acts as a specificity filter that prevents binding of non-NES peptides. The large conformational range of NES backbones explains the lack of a fixed pattern for its 3-5 hydrophobic anchor residues, which in turn explains the large array of peptide sequences that can function as NESs.

*For correspondence: yuhmin.chook@utsouthwestern.edu

**Competing interests:** The authors declare that no competing interests exist.

## Introduction

The chromosome region maintenance 1 protein (CRM1) or Exportin-1 (XPO1) binds 8–15 residues-long nuclear export signals (NESs) in hundreds of different protein cargoes (*Fornerod et al., 1997*; *Fukuda et al., 1997*; *Stade et al., 1997*; *Ossareh-Nazari et al., 1997*; *Xu et al., 2012a*; *Kırlı et al., 2015*; *Thakar et al., 2013*). NES sequences are very diverse but each usually has 4–5 hydrophobic residues (often Leu/Val/Ile/Phe/Met; labeled $\Phi$0–5) that bind hydrophobic pockets (labeled P0–P4) in a hydrophobic groove formed by HEAT repeats 11 and 12 of CRM1 (9–16). The hydrophobic anchor residues are arranged in many ways, currently described by ten consensus patterns for corresponding NES classes 1a, 1b, 1c, 1d, 2, 3, 1a-R, 1b-R, 1c-R and 1d-R (*Figure 1A*) (*Fung et al., 2015*).

Previous structures of CRM1 bound to five different NESs showed virtually identical NES-binding grooves (*Fung et al., 2015*; *Monecke et al., 2009*; *Dong et al., 2009*; *Güttler et al., 2010*). NESs from Snurportin-1 (SNUPN[NES]; class 1c) and Protein Kinase A Inhibitor (PKI[NES]; class 1a) bind CRM1 as $\alpha$-helix followed by a short $\beta$-strand, while the proline-rich NES from HIV-1 REV (Rev[NES]; class 2) adopts mostly extended conformation (*Figure 1B*) (*Güttler et al., 2010*). The majority of CRM1-NES interactions involve NES hydrophobic anchor side chains, with very few polar and main chain interactions. Previously, we studied NESs with the $\Phi1XX\Phi2XXX\Phi3XX\Phi4$ (class 3) pattern where the $i$, $i + 3$, $i + 7$, $i + 10$ $\Phi$ positions suggested a single long amphipathic helix. However, it is perplexing that a long all-helical peptide could fit in the narrow tapering CRM1 groove. Structures of such NESs from kinase RIO2 and cytoplasmic polyadenylation element-binding protein 4 (hRio2[NES], CPEB4[NES]) showed that they do not adopt all-helical conformations but unexpectedly adopt helix-strand conformations that bind CRM1 in the opposite or minus (−) polypeptide direction to that of SNUPN[NES], PKI[NES] and Rev[NES] ((+) NESs) (*Fung et al., 2015*). hRio2[NES] and CPEB4[NES] were hence reclassified as class 1a-R NESs (*Figure 1A*).

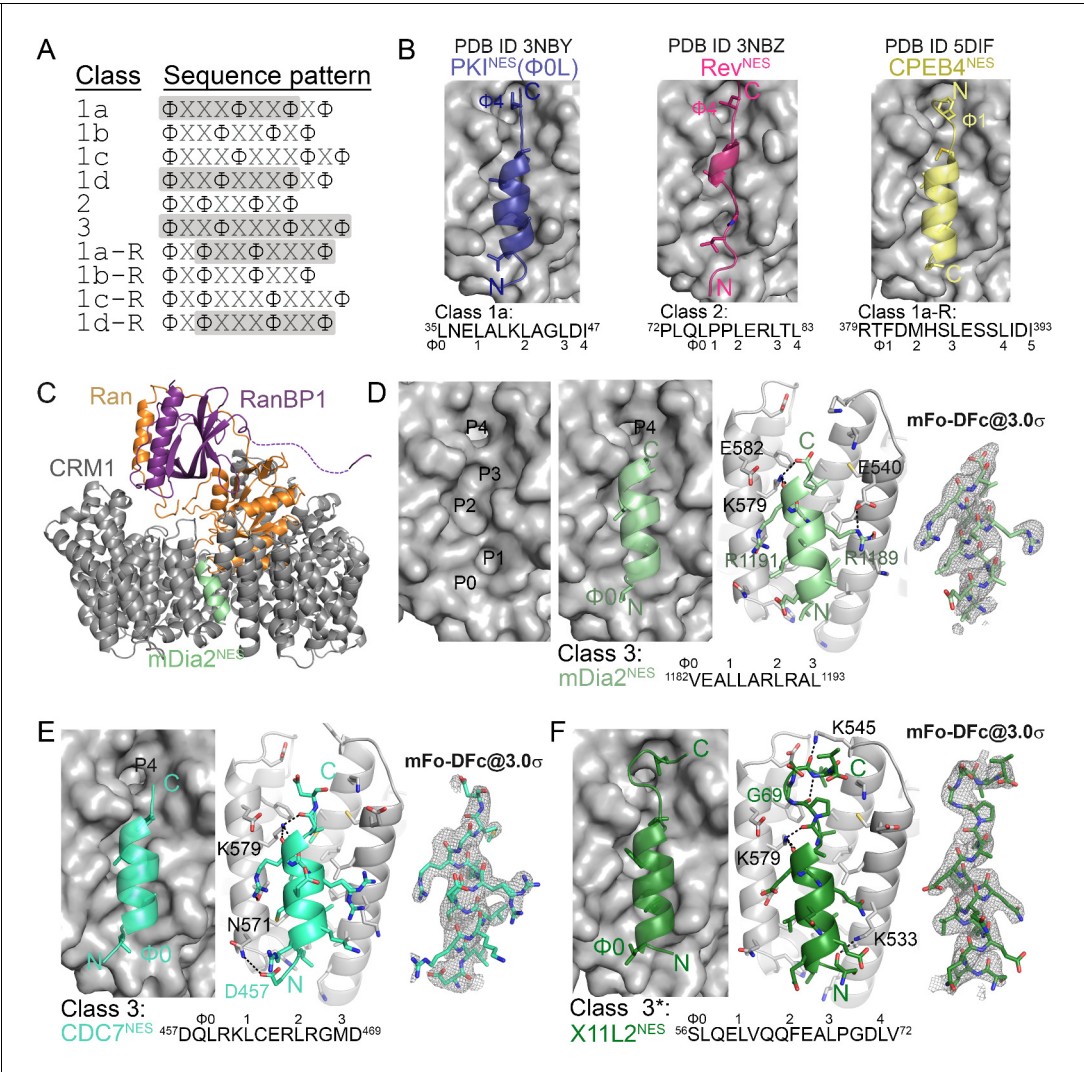

**Figure 1.** Structures of CRM1-bound NESs that match the potentially all-helical class 3 pattern. (**A**) Current NES sequence patterns (Φ is Leu, Val, Ile, Phe or Met and X is any amino acid). Potential amphipathic α-helices, predicted with hydrophobic patterns of $i$, $i + 4$, $i + 7$ or $i$, $i + 3$, $i + 7$ or $i$, $i + 3$, $i + 7$, $i + 10$, are shaded grey. (**B**) Structure of PKI$^{NES}$(Φ0L) (dark blue, PDB ID: 3NBY), Rev$^{NES}$ (pink, 3NBZ) and CPEB4$^{NES}$ (yellow, 5DIF) bound to the NES-binding groove of CRM1 (grey surface). NESs are shown in cartoon representations with their Φ side chains shown as sticks. (**C**) The overall structure of the engineered $^{Sc}$CRM1 (grey)-Ran•GTP (orange)-RanBP1 (purple)-mDia2$^{NES}$ (pale green) complex. The structure of (**D**) mDia2$^{NES}$ (pale green), (**E**) CDC7$^{NES}$ (green-cyan) and (**F**) X11L2$^{NES}$ (forest) bound to the NES-binding groove of $^{Sc}$CRM1 in the engineered $^{Sc}$CRM1-Ran-RanBP1 complex. *The X11L2$^{NES}$ sequence matches the class 3 pattern, but binds CRM1 according to the new hydrophobic pattern Φ0XXΦ1XXXΦ2XXΦ3XXXΦ4 that we termed the class 4 pattern. mDia2$^{NES}$ is not shown in the leftmost panel of (**D**) to view the five hydrophobic pockets (**P0**–**P4**) of the CRM1 groove. Rightmost panels of (**D**–**F**): overlays of 3.0σ positive densities of kick OMIT mFo-DFc maps (calculated with peptides omitted) and final coordinates of the NES peptides. Middle panels of (**D**–**F**): black dashes show CRM1-NES hydrogen bonds and polar contacts.

The following source data and figure supplements are available for figure 1:

**Source data 1.** Data collection and refinement statistics.

**Source data 2.** Data collection and refinement statistics (cont.).

**Source data 3.** Crystallization conditions of CRM1-NES complexes.

**Figure supplement 1.** Binding affinities of NES to CRM1 measured by differential bleaching experiments.

**Figure supplement 2.** Modeling strategy for NESs with weak Φ side chain densities.

*Figure 1 continued on next page*

*Figure 1 continued*

**Figure supplement 3.** Mutagenesis of FMRP-1b[NES] to validate the sequence assignment of the NES.

**Figure supplement 4.** Electron densities of the NES peptides.

**Figure supplement 5.** A longer CDC7[NES] peptide binds the CRM1 groove without interacting with the P4 pocket.

**Figure supplement 6.** Variable number of Φ side chains are needed to have an active NES.

We do not understand the extent of NES structural diversity nor how NESs with different hydrophobic patterns that presumably reflect different secondary structures all bind to the seemingly invariant and three-dimensionally constrained CRM1 groove. Here, eight new structures of CRM1 bound to diverse NESs show several different and unexpected NES backbone conformations that share only a common one-turn helix element. All NESs also participate in hydrogen bonding with Lys568 of $^{Hs}$CRM1, and mutagenic/structural analysis identifies Lys568 as a selectivity filter that blocks binding of non-NES peptides.

## Results

### CRM1-bound NESs adopt diverse conformations

We study three NESs that uniquely match the all-helical class 3 pattern (Φ1XXΦ2XXXΦ3XXΦ4). Because most previously studied NESs have substantial helical content, we also study five NESs that match class 2 (Φ1XΦ2XXΦ3XΦ4) and class 1b (Φ1XXΦ2XXΦ3XΦ4) patterns, where the hydrophobic residue positions do not suggest an amphipathic helix (*Figure 1A*). All eight NESs bind $^{Hs}$CRM1 in the presence of RanGTP with dissociation constants ($K_{Ds}$) of 670 nM-20 μM, and were crystallized bound to the previously described engineered $^{Sc}$CRM1-RanGppNHp-Yrb1p complex (*Figure 1—figure supplement 1*, *Figure 1—source data 1*, *2* and *3*) (*Fung et al., 2015*). Details of how the NES peptides were modelled can be found in methods section and in *Figure 1—figure supplements 2* and *3*. NES-bound CRM1 grooves in the structures (2.1–2.4 Å resolution) resemble the PKI[NES]-bound $^{Mm}$CRM1 groove (Cα/all-atom rmsds 0.5 Å/1.1 Å for 85 groove residues), and all NESs use 4–5 hydrophobic anchor residues to bind P0-P4 hydrophobic pockets of CRM1 (*Figures 1D–F* and *2*) (*Güttler et al., 2010*).

Class 3 NESs from mouse diaphanous homolog 3 (mDia2[NES: 1179]SVPEVEALLARLRAL[1193]) and the cell division cycle 7-related protein kinase (CDC7[NES: 456]QDLRKLCERLRGMDSSTP[473]) are indeed all-helix peptides, both forming 3-turn α-helices that occupy only the wide part of the CRM1 groove (*Figure 1D,E*, *Figure 1—figure supplement 4*). The last residue of the mDia2 protein (Leu1193) binds the CRM1 P3 pocket leaving the P4 pocket empty. CDC7[NES] is far from the protein C-terminus but structures of longer peptides suggest that CDC7[NES] exits the groove after Met468 or Φ3 (*Figure 1—figure supplement 5*). The mDia2[NES] and CDC7[NES] sequence patterns should thus be Φ0XXΦ1XXXΦ2XXΦ3. The Φ4 anchor position is clearly not used in mDia2[NES] and CDC7[NES] even though Φ4 is key for activities of several other NESs (*Wen et al., 1995*; *Meyer et al., 1996*; *Richards et al., 1996*; *Scott et al., 2002*; *Tsukahara and Maru, 2004*). The number of Φ anchor residues necessary for CRM1 binding can vary between 3–5 (*Figure 1—figure supplement 6*). A third class 3-matching NES from beta-amyloid binding protein X11L2 (X11L2[NES: 55]SSLQELVQQFEALPGDLV[72]) binds differently (*Figure 1F*, *Figure 1—figure supplement 4*). [57]LQELVQQFEAL[67] forms a 3-turn α-helix, [68]PGDL[71] forms a type I β-turn, and X11L2[NES] therefore exhibits a new Φ0XXΦ1XXXΦ2XXΦ3XXXΦ4 (class 4) pattern.

Structures of NESs with non-helical patterns are also informative. The previous structure of Rev[NES] (class 2) suggested that its three prolines may constrain against a helical conformation (*Figure 1B*) (*Güttler et al., 2010*). Class 2 NESs in the Mothers against decapentaplegic homolog 4 protein (SMAD4[NES: 133]YERVVSPGIDLSGLTLQ[149]) and the fragile X mental retardation protein (FMRP[NES: 423]YLKEVDQLRLERLQI[437]) have few to no prolines but still bind CRM1 with mostly loop-like

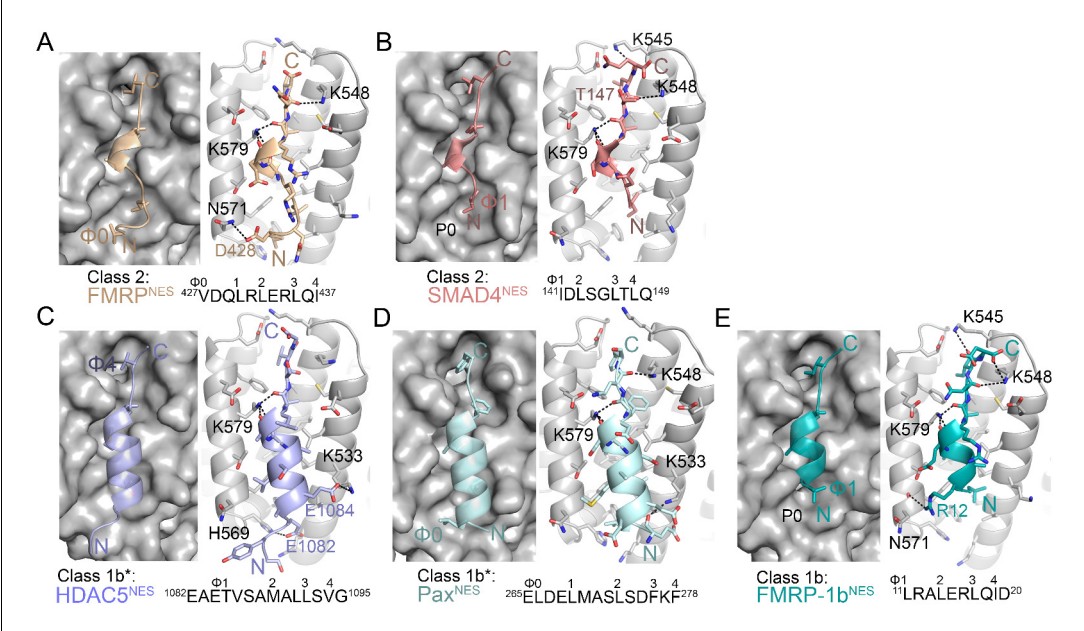

**Figure 2.** Structures of NESs with non-helical sequence patterns. (**A–E**) FMRP[NES] (light orange), SMAD4[NES] (salmon), HDAC[NES] (slate), Pax[NES] (pale cyan) and FMRP-1b[NES] (deep teal) bound to the [Sc]CRM1 groove. Black dashes show CRM1-NES hydrogen bonds and polar contacts, and unoccupied CRM1 hydrophobic pockets are labeled. *HDAC5[NES] and Pax[NES] sequences match the class 1b pattern, but both peptides bind CRM1 using $\Phi$ residues that match class 1a pattern. Average displacements of each $\Phi$ C$\alpha$ in the eight NESs (including mDia2[NES], CDC7[NES], X11L2[NES] in *Figure 1*) from the equivalent $\Phi$ C$\alpha$ of PKI[NES]($\Phi$0L) are 1.3 ± 0.6 ($\Phi$4), 0.8 ± 0.5 ($\Phi$3), 0.7 ± 0.4 ($\Phi$2), 0.9 ± 0.3 ($\Phi$1) and 1.8 ± 0.9 ($\Phi$0) Å.

The following figure supplements are available for figure 2:

**Figure supplement 1.** Electron densities of the NES peptides.

**Figure supplement 2.** Engineering class 1b NESs.

structures (*Figure 2A,B*, *Figure 2—figure supplement 1*). Histone deacetylase 5 (HDAC5[NES: 1082]EAETVSAMALLSVG[1095]) and Paxillin (Pax[NES: 264]RELDELMASLSDFKFMAQ[281]) have NESs with non-α-helical class 1b pattern ($\Phi$1XX$\Phi$2XX$\Phi$3X$\Phi$4), but the peptides bind according to the class 1a pattern instead (*Figure 2C,D*, *Figure 2—figure supplement 1*). This left no other experimentally verified NES in the databases that unambiguously match class 1b pattern (*Kosugi et al., 2008*; *Xu et al., 2012c*). We engineered a class 1b NES by adding an alanine to FMRP[NES] (FMRP-1b[NES]; YLKEVDQLRALERLQID), which forms a short 1.5-turn 3$_{10}$ helix followed by a 3-residue $\beta$-strand (*Figure 2E*, *Figure 2—figure supplements 1* and *2*). The 3$_{10}$ helix favorably presents $\Phi$1 and $\Phi$2 into the CRM1 groove and natural class 1b NESs are likely to bind similarly.

## Structural requirements for an NES

Structures of >13 different CRM1-bound NESs are now available, and may be sorted into five or six groups according to peptide backbone conformations (*Figure 3A*). Class 1 NESs are helix-strand peptides with either α-helices (class 1a, 1c) or 3$_{10}$ helices (class 1b). Class 1-R NESs are strand-helix peptides, class 2 NESs are mostly loop-like and class 3 NESs are all-helix peptides. The helix-$\beta$-turn X11L2[NES] structure revealed a new $\Phi$0XX$\Phi$1XXX$\Phi$2XX$\Phi$3XXX$\Phi$4 (class 4) pattern.

The only common secondary structural element in the NES structures is one turn of NES helix at $\Phi$2X$_{2-3}$$\Phi$3 (grey box, *Figure 3A*). This conserved turn of helix is flanked on one side by additional turns of helix (classes 1, 1-R) or by loops (class 2), and on the other side by $\beta$-strands (classes 1, 1-R, 2) or $\beta$-turn (class 4), or the helix ends as the chain terminates or exits the groove (class 3) (*Figure 3A*). Dihedral (psi) angles in the 1-turn of helix gradually increase in progression from helical to $\beta$-strand conformations (*Figure 3B*).

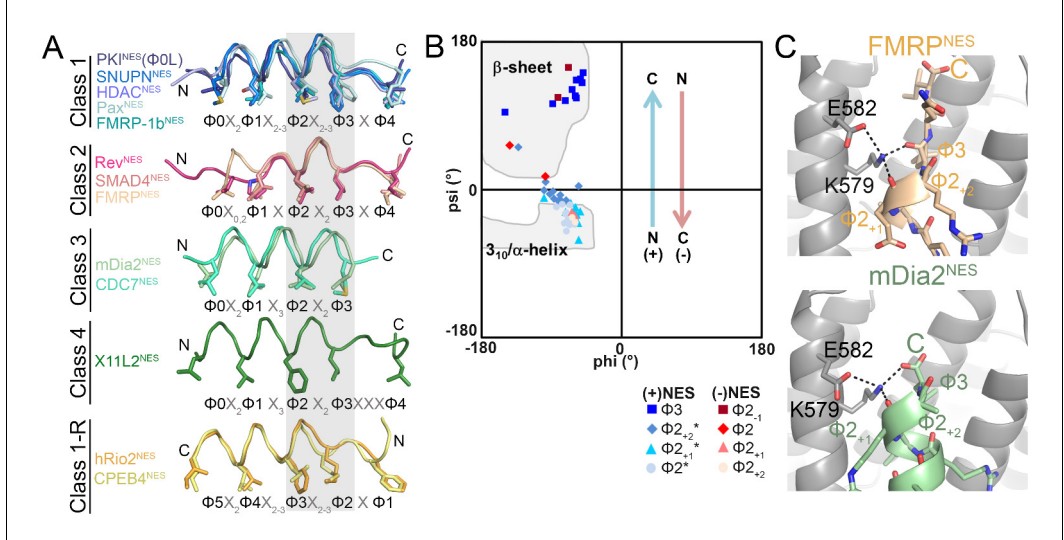

**Figure 3.** NESs adopt diverse conformations to bind CRM1, sharing only a small one turn of the helix structural element. (**A**) Overlay of 13 different CRM1-bound NESs: mDia2[NES], CDC7[NES], X11L2[NES], FMRP[NES], SMAD4[NES], HDAC[NES], Pax[NES] and FMRP-1b[NES], SNUPN[NES] (3GB8), PKI[NES]($\Phi$0L) (3NBY), Rev[NES] (3NBZ), hRio2[NES] (5DHF), CPEB4[NES] (5DIF) (shown as ribbons with $\Phi$ residues in sticks; their CRM1 H12A helices were superimposed). A grey rectangle highlights the only secondary structural element shared by all 13 NESs: a single turn of helix. (**B**) Ramachandran plot of phi/psi angles of the four residues in each of the conserved one-turn of helix. Arrows indicate changes in psi angles along the polypeptide direction. For example, (+) HDAC5[NES]: $\Phi$2 $\Psi$ = −43.5°, $\Phi$2$_{+1}\Psi$ = −21.7°, $\Phi$2$_{+2}\Psi$ = −1.3°, $\Phi$3 $\Psi$ = 133.8° and (−) CPEB4[NES]: $\Phi$2$_{+2}\Psi$ = −40.2°, $\Phi$2$_{+1}\Psi$ = −31.0°, $\Phi$2 $\Psi$ = 16.5°, $\Phi$2$_{-1}\Psi$ = 111.5°. *Residues in SNUPN[NES] plotted are $\Phi$2$_{+1}$, $\Phi$2$_{+2}$, $\Phi$2$_{+3}$ and $\Phi$3. (**C**) Detailed view of niche motifs in (+) NESs, FMRP[NES] and mDia2[NES].

The following figure supplements are available for figure 3:

**Figure supplement 1.** NES main chain hydrogen bonds with $^{Sc}$CRM1 Lys579 in (−) NESs.

**Figure supplement 2.** The Lys568 of $^{Hs}$CRM1 is important for NES binding.

In all (+) NESs, main chain carbonyls of $\Phi$2$_{+1}$ and $\Phi$3 residues in the 1-turn helix element hydrogen bonds with the $^{Sc}$CRM1 Lys579 (or $^{Mm}$CRM1/$^{Hs}$CRM1 Lys568) side chain, much like niche3/4 motifs where carbonyls of residues $i$ and $i$ + 2 or $i$ + 3 coordinate a cationic group (*Torrance et al., 2009*). The $\Phi$3-Lys579 hydrogen bond is possible only because the $\beta$-strand psi angle turns $\Phi$3 carbonyl towards Lys579 (*Figure 3C*). NES helix-Lys579 hydrogen bonds are absent in (−) NESs as backbone carbonyls point in the opposite direction. Here, carbonyls of the N-terminal $\beta$-strand hydrogen bond with Lys579 (*Figure 3—figure supplement 1*). Therefore, another common structural feature of NESs is hydrogen bonding between NES backbone and $^{Sc}$CRM1 Lys579 ($^{Hs}$CRM1 Lys568). Mutations of $^{Hs}$CRM1 Lys568 impair NES binding. Mutants $^{Hs}$CRM1(K568A) and $^{Hs}$CRM1 (K568M) bind FITC-PKI[NES] two to three orders of magnitude weaker than wild type $^{Hs}$CRM1, supporting the importance of Lys568-NES interactions (*Figure 3—figure supplement 2*).

In summary, an active NES (1) can use many different backbone conformations to present 3–5 hydrophobic anchor residues into 3–5 CRM1 hydrophobic pockets (P0 and/or P4 are sometimes not used), (2) has one turn of helix with helix-strand transition that binds the central portion of the CRM1 groove and (3) has backbone conformation that can hydrogen bond with Lys568 of $^{Hs}$CRM1.

## $^{Hs}$CRM1 Lys568 is a selectivity filter for NES recognition

What then are CRM1 groove features that selectively recognize the key NES features? Arrangement of hydrophobic pockets in the groove likely selects NESs with suitably placed $\Phi$ residues. Groove shape, tapering and most constricted at $^{Sc}$CRM1 Lys579 ($^{Hs}$CRM1 Lys568), likely selects for NES

helices that transition to strands or NES helices that end (*Figure 3C*). Is groove-constricting [Hs]CRM1 Lys568 perhaps key for differentiating active from false positive NES sequences? We tested mutants [Hs]CRM1(K568A) and [Hs]CRM1(K568M) for interactions with three previously identified false positive NESs that match NES consensus but do not bind CRM1: peptides from Hexokinase-2 (Hxk2[pep]: [18]DVPKELMQQIENFEKIFTV[36], class 3 match), Deformed Epidermal Autoregulatory Factor 1 homolog (DEAF1[pep:452]SWLYLEEMVNSLLNTAQQ[469]; class 1a-R match) and COMM domain-containing protein 1 (COMMD1[pep; 173]KTLSEVEESISTLISQPN[190]; class 3 match) (*Figure 4A*) (*Xu et al., 2012c*, *2015*). Wild type [Hs]CRM1 does not bind the peptides but [Hs]CRM1(K568A) binds Hxk2[pep] and DEAF1[pep], and [Hs]CRM1(K568M) binds DEAF1[pep] but not Hxk2[pep], suggesting that Lys568 is important in filtering out false positive NESs (*Figure 4A*).

Both [Sc]CRM1(K579A)-bound Hxk2[pep] and DEAF1[pep] are all-helix peptides (*Figure 4B,C*, *Figure 4—figure supplement 1*, *Figure 4—source data 1*). The fourth turn of the Hxk2[pep] helix packs into hydrophobic space widened by removal and rearrangement of the Lys579 and Glu582 side chains, respectively (*Figure 4B*). The 2.5-turn α-helix of DEAF1[pep]binds in the (−) direction and is slightly longer than helices in true (−) NESs (*Figure 4C*). Superpositions of Hxk2[pep] and DEAF1[pep] onto wild type CRM1 grooves show the fourth turn of the Hxk2[pep] helix and the N-terminus of the DEAF1[pep] helix clashing with [Sc]CRM1 Lys579/[Mm]CRM1 Lys568 side chains (*Figure 4B,C*, *Figure 4—figure supplement 2*). The rest of the mutant [Sc]CRM1(K579A) groove is highly similar to the wild type groove. Therefore, the key feature of the wild type groove that prevents Hxk2[pep] and DEAF1[pep] binding is Lys568, which is not only a critical hydrogen bond donor for binding NESs, but its long side chain also blocks binding of sequences that do not meet NES structural requirements.

## Discussions

Class 1a, 1b, 1c, 2, 3, 4 and 1a-R NES structures show 5–6 distinct backbone conformations that match their respective hydrophobic sequence patterns. We can infer structures of remaining NES classes: class 1d NESs ($\Phi1XX\Phi2XXX\Phi3X\Phi4$) are likely α-helix-strand and other (−) NES classes are likely the reverse of their (+) counterparts. Symmetrical class 2, 3 and 4 patterns are identical in both (+)/(−) directions but (−) class 3 and 4 NESs, lacking β-strands to hydrogen bond with [Hs]CRM1 Lys568, may not be ideal NESs.

Structures of many diverse NES sequences suggest how one unchanging peptide-bound CRM1 groove can recognize up to a thousand different peptides. Dependence of 3–5 hydrophobic residues in 8–15 residues-long NES arises from the substantial binding energy of anchor hydrophobic side chains interacting with 3–5 CRM1 hydrophobic pockets. However, lack of contact with NES backbone allows anchor side chains to be presented in many conformations including both N- to C-terminal orientations, explaining broad specificity defined by highly variable spacings between anchors. Interestingly, NES conformation is not entirely unrestrained, as CRM1 groove constriction imposes either exit/termination of the NES chain or its continuation in extended configuration. Solutions for the broadly specific NES recognition contrast with those of analogous systems. MHC I and II proteins, each recognizing at least hundreds of different peptide antigens, use many peptide main chain contacts for affinity with only a few supplementary peptide side chain interactions (*Zhang et al., 1998*; *Madden, 1995*). The result here is a conformational selection of particular lengths of extended peptides binding in conserved N- to C-terminal orientation, with little sequence restriction. The Calmodulin-helical peptide system is yet another contrast, as the binding domain uses its flexible fold to adapt to various helical ligands (*Tidow and Nissen, 2013*; *Hoeflich and Ikura, 2002*).

CRM1-NES structures expanded the six NES patterns derived from peptide library studies to the eleven patterns shown in *Figures 1A* and *3A*. The ever-expanding set of NES patterns suggests that no fixed hydrophobic pattern likely describes the NES. Furthermore, only ~50% of consensus-matching previously reported NESs that were tested actually bound CRM1, contributing to the inefficiency of available NES predictors (with precision of 50% at 20% recall rate) (*Xu et al., 2012b*, *2015*; *Kosugi et al., 2014*; *Fu et al., 2011*). The many available NES structures, diversity of NES conformations and the structurally conserved one-turn helix NES element revealed here will enable development of structure- rather than sequence-based NES predictors (*Raveh et al., 2011*; *Schindler et al., 2015*; *Trellet et al., 2013*; *Yan et al., 2016*). There is a need to identify many more CRM1 cargoes as apoptosis of different cancer cells upon CRM1 inhibition by the drug Selinexor (in clinical trials for a variety of cancers) and other inhibitors (*Parikh et al., 2014*; *Mendonca et al.,*

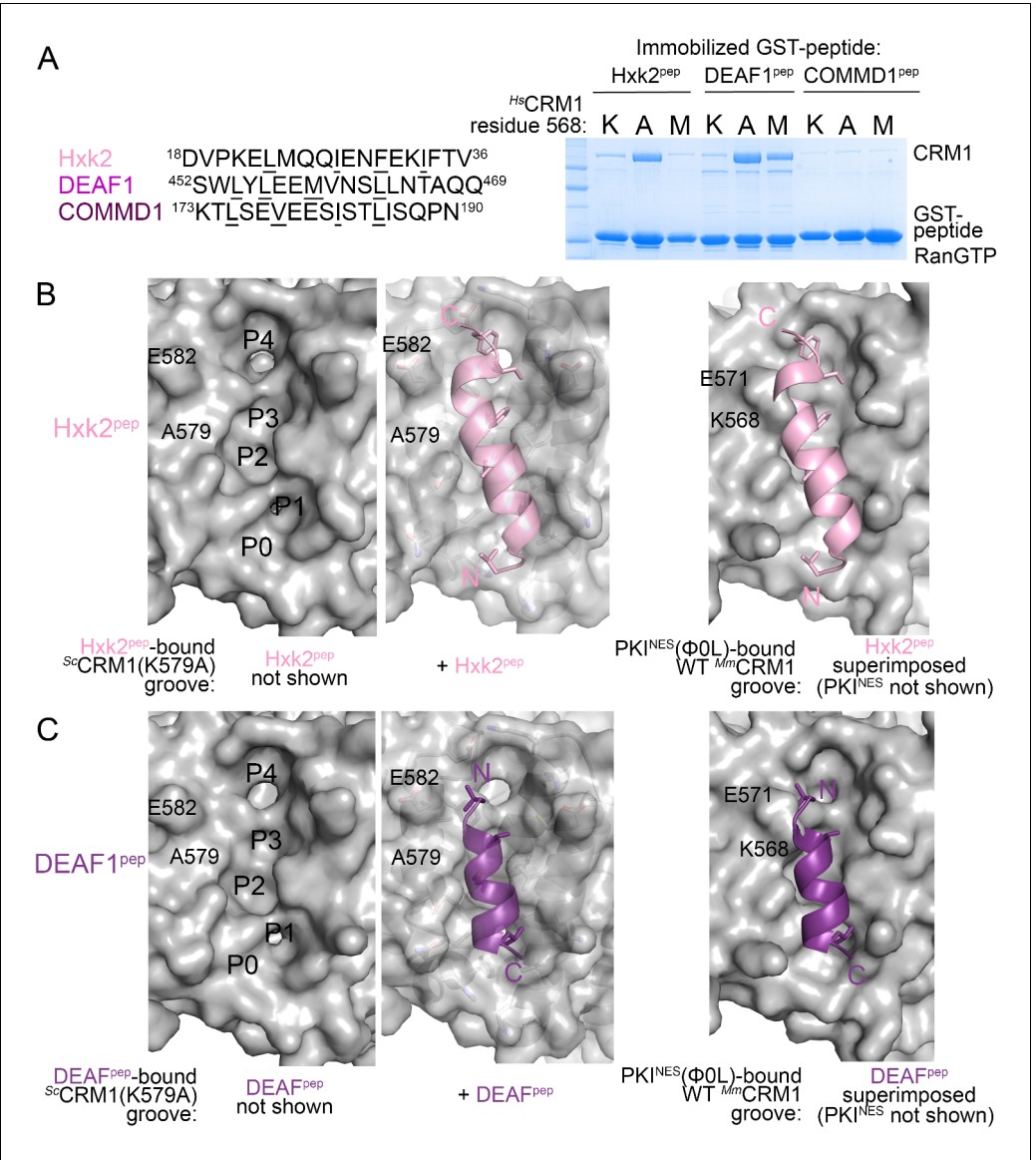

**Figure 4.** [Hs]CRM1 Lys568 is a selectivity filter. (**A**) False positive NES sequences with Φ residues of consensus matches underlined. Pull-down binding assay of ~5 μg immobilized GST-NESs and 7.5 μM [Sc]RanGTP with 2.5 μM of either wild type [Hs]CRM1 or mutant [Hs]CRM1(K568A) or [Hs]CRM1(K568M) in 200 μL reactions. (**C–D**) Structures of Hxk2[pep] (pink) (**C**) and DEAF1[pep] (purple) (**D**) bound to [Sc]CRM1(K579A). Left panels, peptides were removed to show the surface of the mutant [Sc]CRM1(K579A) groove. Middle panels, peptides bound to the mutant [Sc]CRM1 (K579A). Right panels, CRM1(K579A)-bound Hxk2[pep] and DEAF1[pep] superimposed onto the PKI[NES](Φ0L)-bound [Mm]CRM1 groove (3NBY; CRM1 H12A helices were aligned; PKI[NES] not shown) to show steric clash of the Hxk2[pep] and DEAF1[pep] peptides with the [Mm]CRM1 Lys568 side chain.

The following source data and figure supplements are available for figure 4:

**Source data 1.** Data collection and refinement statistics and crystallization conditions.

**Figure supplement 1.** Electron densities of the HXK2[pep] and DEAF[pep] false positive NES peptides bound to the [Sc]CRM1(K579A) mutant.

**Figure supplement 2.** Additional overlays of Hxk2[pep] and DEAF1[pep] onto NES-bound wild type CRM1 grooves.

*2014*; *Das et al., 2015*; *Alexander et al., 2016*; *Gounder et al., 2016*; *Abdul Razak et al., 2016*; *Lapalombella et al., 2012*; *Inoue et al., 2013*; *Etchin et al., 2013*; *Tai et al., 2014*; *Cheng et al., 2014*; *Kim et al., 2016*; *Hing et al., 2016*; *Vercruysse et al., 2016*) appears to be driven by nuclear accumulation of different sets of NES-containing cargoes, but identities of most of these apoptosis-causing cargoes are still unknown.

Finally, we find that the [Hs]CRM1 Lys568 side chain acts as a filter that physically selects for NESs with helices that transition to strands or end at the narrow part of the CRM1 groove. Interestingly, Lys568 interacts electrostatically with [Hs]CRM1 Glu571, which is mutated to glycine or lysine in chronic lymphocytic leukemia and lymphomas with poor prognosis (*Puente et al., 2011*; *Jardin et al., 2016*; *Camus et al., 2016*). Disease mutations will abolish Glu571-Lys568 contacts and possibly affect NES binding and selectivity.

## Materials and methods

### Crystallization of CRM1-Ran-RanBP1-NES complexes

CRM1 ([Sc]CRM1 residues 1–1058, △377–413, [537]DLTVK[541] to GLCEQ, V441D), Yrb1p (yeast RanBP1 residues 62–201), human Ran full length and various NESs were expressed and purified as described in Fung et al. (*Fung et al., 2015*). CRM1 (K579A) mutant was expressed in pGex-TEV and purified like CRM1. Crystallization, data collection and processing, and solving of the structures were also performed in the same manner as previously described. X-ray/stereochemistry weight and X-ray/ADP weight were both optimized by phenix.refine in PHENIX (RRID:SCR_014224).

NES peptides are modeled according to the positive difference density (2mFo-DFc map) at the binding groove after refinement of the CRM1-Ran-RanBP1 model. In all structures, there are good electron densities for the NES main chain and directions of side chain density in the helical portion of the peptides allow unambiguous determination that they are all oriented in the positive (+) NES orientation. Side chain assignments of the NES peptides are guided by (1) densities of Φ side chains that point into the binding groove, (2) densities for long non-Φ side chains such as arginine, phenylalanine and methionine and (3) physical considerations such as steric clashes.

For example, to model the bound mDia2[NES] peptide (sequence: GGSY-[1179]SVPEVEALLARLRA L[1193]), we made use of the obvious electron densities (mFo-DFc map) for long side chains to guide sequence assignment. There is a strong side chain density suitable for an arginine side chain on the peptide (white dashed circle in *Figure 1—figure supplement 2A*). There are only two arginine residues in the mDia2[NES] peptide, Arg1189 and Arg1191. If the long side chain density is assigned to Arg1189, then Arg1191 would end up pointing into the binding groove – a very energetically unfavorable and unlikely situation. Furthermore, Ala1188 would end up in the P2 pocket of CRM1 where there is an obvious density for a longer hydrophobic side chain (left panel, *Figure 1—figure supplement 2A*). On the other hand, when Arg1191 is assigned to the long and continuous side chain density (adjacent to helix H12A of CRM1), the remaining side chains in the NES end up in positions that are consistent the electron densities.

For the FMRP-1b[NES] (sequence: [1]GGS-YLKEVDQLRALERLQID[20]), there are no obvious long side chain densities that could help with modeling. There are however obvious densities for several side chains in the first two turn of the NES helix. These side chain densities are consistent with two possible sequence assignments: [5]LKEVDQLRAL[14] or the more C-terminal [11]LRALERLQID[20]. We tested modeling of FMRP-1b[NES] by refining both peptide models and by testing a mutant peptide that should distinguish between the two models. Ten cycles of PHENIX refinement of the [5]LKEVDQL-RAL[14] model resulted in positive and negative difference densities (mFo-DFc map) at several NES side chains, which suggested an incorrect assignment (left panels, *Figure 1—figure supplement 2B*). In contrast, different densities are absent when the [11]LRALERLQID[20] model is refined (right panels, *Figure 1—figure supplement 2B*). The final FMRP-1b[NES] structure was therefore modelled as [11]LRALERLQID[20]. The sequence assignment of FMRP-1b[NES] was also tested using a mutant FMRP-1b[NES] that has the sequence YLKEVDQLRALER. If the NES is [5]LKEVDQLRAL[14], FMRP-1b[NES] mutant YLKEVDQLRALER should bind well to CRM1. However, if [11]LRALERLQID[20] is the FMRP-1b[NES], mutant YLKEVDQLRALER should not bind CRM1 as the C-terminal half of the NES or [17]LQID[20] which includes Φ3 and Φ4 is missing. Results in *Figure 1—figure supplement 3* show that FMRP-1b[NES]

mutant YLKEVDQLRALER does not bind CRM1, providing further support that the NES is indeed [11]LRALERLQID[20] as currently assigned.

## NES activity assays

Pull-down binding assays, in vivo NES activity assay and differential bleaching experiments for determining binding affinities were all performed the same way as described in *Fung et al. (2015)*. The data were analyzed in PALMIST (*Scheuermann et al., 2016*) and plotted with GUSSI (*Brautigam, 2015*).

## Accession codes

Structures and crystallographic data have been deposited at the PDB: 5UWI (CRM1-HDAC5[NES]), 5UWH (CRM1-Pax[NES]), 5UWO (CRM1-FMRP-1b[NES]), 5UWJ (CRM1-FMRP[NES]), 5UWU (CRM1-SMAD4[NES]) 5UWP (CRM1-mDia2[NES]), 5UWQ (CRM1-CDC7[NES]), 5UWR (CRM1-CDC7[NES] ext), 5UWS (CRM1-X11L2[NES]), 5UWT (CRM1(K579A)-Hxk2[pep]), 5UWW (CRM1(K579A)-DEAF1[pep]).

## Acknowledgements

We thank the Structural Biology Laboratory and Macromolecular Biophysics Resource at UTSW for their assistance with crystallographic and biophysical data collection. Results shown in this report are derived from work performed at Argonne National Laboratory, Structural Biology Center at the Advanced Photon Source. Argonne is operated by UChicago Argonne, LLC, for the US Department of Energy, Office of Biological and Environmental Research under contract DE-AC02-06CH11357. Ho Yee Joyce Fung is a Howard Hughes Medical Institute International Student Research fellow. This work was funded by the Cancer Prevention Research Institute of Texas (CPRIT) Grants RP120352 and RP150053 (YMC), R01 GM069909 (YMC), the Welch Foundation Grant I-1532 (YMC), Leukemia and Lymphoma Society Scholar Award (YMC), the University of Texas Southwestern Endowed Scholars Program (YMC), and a Croucher Foundation Scholarship (HYJF).

## Additional information

### Funding

| Funder | Grant reference number | Author |
|---|---|---|
| Howard Hughes Medical Institute | International Student Research fellow | Ho Yee Joyce Fung |
| Croucher Foundation | Graduate Student Scholarship | Ho Yee Joyce Fung |
| Cancer Prevention and Research Institute of Texas | RP120352 | Yuh Min Chook |
| Cancer Prevention and Research Institute of Texas | RP150053 | Yuh Min Chook |
| National Institutes of Health | R01 GM069909 | Yuh Min Chook |
| Welch Foundation | I-1532 | Yuh Min Chook |
| Leukemia and Lymphoma Society | Scholar Award | Yuh Min Chook |
| University of Texas Southwestern Medical Center | Endowed Scholars Program | Yuh Min Chook |

The funders had no role in study design, data collection and interpretation, or the decision to submit the work for publication.

### Author contributions

HYJF, Conceptualization, Formal analysis, Funding acquisition, Validation, Investigation, Visualization, Methodology, Writing—original draft, Writing—review and editing; S-CF, Formal analysis, Investigation, Visualization, Methodology; YMC, Conceptualization, Supervision, Funding acquisition, Visualization, Methodology, Writing—original draft, Writing—review and editing

Author ORCIDs
Ho Yee Joyce Fung, http://orcid.org/0000-0002-0502-1957
Yuh Min Chook, http://orcid.org/0000-0002-4974-0726

# Additional files

## Major datasets

The following datasets were generated:

| Author(s) | Year | Dataset title | Dataset URL | Database, license, and accessibility information |
|---|---|---|---|---|
| Ho Yee Joyce Fung, Yuh Min Chook | 2017 | Crystal Structure of HDAC5 NES Peptide in complex with CRM1-Ran-RanBP1 | http://www.rcsb.org/pdb/explore/explore.do?structureId=5UWI | Publicly available at the RCSB Protein Data Bank (accession no. 5UWI) |
| Ho Yee Joyce Fung, Yuh Min Chook | 2017 | Crystal Structure of Paxillin NES Peptide in complex with CRM1-Ran-RanBP1 | www.rcsb.org/pdb/explore/explore.do?structureId=5UWH | Publicly available at the RCSB Protein Data Bank (accession no. 5UWH) |
| Ho Yee Joyce Fung, Yuh Min Chook | 2017 | Crystal Structure of Engineered FMRP-1b NES Peptide in complex with CRM1-Ran-RanBP1 | www.rcsb.org/pdb/explore/explore.do?structureId=5UWO | Publicly available at the RCSB Protein Data Bank (accession no. 5UWO) |
| Ho Yee Joyce Fung, Yuh Min Chook | 2017 | Crystal Structure of FMRP NES Peptide in complex with CRM1-Ran-RanBP1 | www.rcsb.org/pdb/explore/explore.do?structureId=5UWJ | Publicly available at the RCSB Protein Data Bank (accession no. 5UWJ) |
| Ho Yee Joyce Fung, Yuh Min Chook | 2017 | Crystal Structure of SMAD4 NES Peptide in complex with CRM1-Ran-RanBP1 | www.rcsb.org/pdb/explore/explore.do?structureId=5UWU | Publicly available at the RCSB Protein Data Bank (accession no. 5UWU) |
| Ho Yee Joyce Fung, Yuh Min Chook | 2017 | Crystal Structure of mDia2 NES Peptide in complex with CRM1-Ran-RanBP1 | www.rcsb.org/pdb/explore/explore.do?structureId=5UWP | Publicly available at the RCSB Protein Data Bank (accession no. 5UWP) |
| Ho Yee Joyce Fung, Yuh Min Chook | 2017 | Crystal Structure of CDC7 NES Peptide in complex with CRM1-Ran-RanBP1 | www.rcsb.org/pdb/explore/explore.do?structureId=5UWQ | Publicly available at the RCSB Protein Data Bank (accession no. 5UWQ) |
| Ho Yee Joyce Fung, Yuh Min Chook | 2017 | Crystal Structure of CDC7 NES Peptide (extended) in complex with CRM1-Ran-RanBP1 | www.rcsb.org/pdb/explore/explore.do?structureId=5UWR | Publicly available at the RCSB Protein Data Bank (accession no. 5UWR) |
| Ho Yee Joyce Fung, Yuh Min Chook | 2017 | Crystal Structure of X11L2 NES Peptide in complex with CRM1-Ran-RanBP1 | www.rcsb.org/pdb/explore/explore.do?structureId=5UWS | Publicly available at the RCSB Protein Data Bank (accession no. 5UWS) |
| Ho Yee Joyce Fung, Yuh Min Chook | 2017 | Crystal Structure of Hxk2 Peptide in complex with CRM1 K579A mutant-Ran-RanBP1 | www.rcsb.org/pdb/explore/explore.do?structureId=5UWT | Publicly available at the RCSB Protein Data Bank (accession no. 5UWT) |
| Ho Yee Joyce Fung, Yuh Min Chook | 2017 | Crystal Structure of DEAF1 Peptide in complex with CRM1 K579A mutant-Ran-RanBP1 | www.rcsb.org/pdb/explore/explore.do?structureId=5UWW | Publicly available at the RCSB Protein Data Bank (accession no. 5UWW) |

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
