## [Decision Letter]

Thank you for submitting your article "CRM1 recognizes diverse Nuclear Export Signals conformations" for consideration by *eLife*. Your article has been reviewed by two peer reviewers, and the evaluation has been overseen by a Reviewing Editor and John Kuriyan as the Senior Editor. The following individuals involved in review of your submission have agreed to reveal their identity: Gino Cingolani (Reviewer #1) and Douglas Barrick (Reviewer #2).

The reviewers have discussed the reviews with one another and the Reviewing Editor has drafted this decision to help you prepare a revised submission.

Summary:

This "Research Advances" manuscript describes continued structural analysis of NES sequences bound to CRM1, a nuclear export receptor. In the previous 2015 *eLife* paper, the authors published two crystal structures of bound sequences and showed that they can interact in different ways, reversing chain orientation. In the present study, they solve eight additional structures from diverse sequences, and characterize additional variation in the mode of binding. The results demonstrate that a 'folded' secondary structure is not a prerequisite to bear a functional NES. Instead, sidechains from the NES sequences arranged in conformations that match the tight and restrained groove generated at the HEAT:HEAT interface are the key determinants. The results help to define different sequence motifs for binding, and how they translate into structure. The present study is certainly an advance from the previous work. As such, the manuscript is favorable for the format of "Research Advances" in *eLife*.

Essential revisions:

1) One major comment to be addressed in revision is that the authors should do a better job in Figure 1 showing the sequences in each NES subfamily, their expected structures, and their actual structures (perhaps a string of h's, s's, and t's). And importantly, indicate which sequences are from the current study, and which are from previous studies. Without a better guide, most readers won't remember which NES sequence has which structure, sequence motif, and orientation.

2) Although all of the crystal structures are solved at a resolution ranging from 2.0~2.5 Å, the electron densities for some of the NES peptides may not have sufficient quality for reliably model building (some of the sidechain densities for the critical hydrophobic residues are missing, e.g. FMRP-1b, mDia-2, etc.). The overall statistics (R factor and R-free factor) cannot reflect this because the peptide only constitutes a small portion of the entire structure. Moreover, the B factors for some of the peptides are extremely high compared to the CRM1*-Ran-RanBP1 protein, suggesting overfitting of the peptides to some extent. The authors need to address this point by stating the rationales for modelling the peptides as they reported.

---

## [Author Response]

*Essential revisions:*

*1) One major comment to be addressed in revision is that the authors should do a better job in Figure 1 showing the sequences in each NES subfamily, their expected structures, and their actual structures (perhaps a string of h's, s's, and t's). And importantly, indicate which sequences are from the current study, and which are from previous studies. Without a better guide, most readers won't remember which NES sequence has which structure, sequence motif, and orientation.*

To better show the sequence patterns of each NES subfamily, we changed the font in Figure 1 of the manuscript to the monospace Courier New font. To incorporate information about potential α-helices within consensus sequences of the different NES classes or subfamily, we shaded these regions grey. These regions of the NESs are predicted to be amphipathic α-helices because their hydrophobic residues form patterns of *i, i*+4, *i*+7 or *i, i*+3, *i*+7 or *i, i*+3, *i*+7, *i*+10. This explanation of how we predict α-helical parts of NESs is now included in 1) the legend for Figure 1, Figure 2) the revised Introduction where we added new text to better explain the unexpected previously solved structures of CRM1-bound hRio2^NES^ and CPEB4^NES^ which has the potentially all-helical class 3 pattern, and 3) the Results section where we revised the text to include rationale for the potentially non-helical class 2 and class 1b NESs.

We also added information into Figure 1 and Figure 2 of the manuscript so that they can serve as better guides to clearly distinguish previous structures from new structures, and connect sequence motifs with peptide structure/orientation:

1) We added prominently displayed PDB IDs to the structures in Figure 1 of the manuscript to clearly indicate that all three peptides are previously published structures.

2) In Figure 1, we also added the sequence of each NES and their respective NES class, marked Φ0-Φ4 residues in the sequences and labeled one Φ residue of each peptide structure to connect the structures to the sequences.

3) In Figure 1 and Figure 2, we clearly indicated the NES class or subfamily for each NES peptide, marked the Φ0-Φ4 residues in the sequences, and one Φ residue in each peptide structure.

Changes in the text of the manuscript:

We revised the legend for Figure 1 according to the changes above: “(A) NES sequence patterns (Φ is Leu, Val, Ile, Phe or Met and X is any amino acid). Potential amphipathic α-helices, predicted with hydrophobic patterns of *i, i*+4, *i*+7 or *i, i*+3, *i*+7 or *i, i*+3, *i*+7, *i*+10, are shaded grey.”

We also revised text in the Introduction and Results of the manuscript. The revised text in Introduction reads: “Previously, we studied NESs with the Φ1XXΦ2XXXΦ3XXΦ4 (class 3) pattern where the *i, i*+3, *i*+7, *i*+10 Φ positions suggested a single long amphipathic helix. […] Structures of such NESs from kinase RIO2 and cytoplasmic polyadenylation element-binding protein 4 (hRio2^NES^, CPEB4^NES^) showed that they do not adopt all-helical conformations but unexpectedly adopt helix-strand conformations that bind CRM1 in the opposite or minus (-) polypeptide direction to that of SNUPN^NES^, PKI^NES^ and Rev^NES^ ((+) NESs) (16).”

The revised text in Results reads: “We study three NESs that uniquely match the all-helical class 3 pattern (Ф1XXФ2XXXФ3XXФ4). Because most previously studied NESs have substantial helical content, we also study five NESs that match class 2 (Ф1XФ2XXФ3XФ4) and class 1b (Ф1XXФ2XXФ3XФ4) patterns, where the hydrophobic residue positions do not suggest an amphipathic helix (Figure 1).”

*2) Although all of the crystal structures are solved at a resolution ranging from 2.0~2.5 Å, the electron densities for some of the NES peptides may not have sufficient quality for reliably model building (some of the sidechain densities for the critical hydrophobic residues are missing, e.g. FMRP-1b, mDia-2, etc.). The overall statistics (R factor and R-free factor) cannot reflect this because the peptide only constitutes a small portion of the entire structure. Moreover, the B factors for some of the peptides are extremely high compared to the CRM1*-Ran-RanBP1 protein, suggesting overfitting of the peptides to some extent. The authors need to address this point by stating the rationales for modelling the peptides as they reported.*

We thank the reviewers for pointing out the need to provide clear rationale for the way we modeled the NES peptides. As they noted, densities are missing for the Φ1 and Φ3 side chains of mDia2^NES^ and for the Φ1 side chain of FMRP-1b^NES^.

To model the bound mDia2^NES^ peptide (sequence: GGSY-^1179^SVPEVEALLARLRAL^1193^), we made use of the obvious electron densities (mFo-DFc map) for long side chains to guide sequence assignment. There is strong side chain density suitable for an arginine side chain on the peptide (white dashed circle in Figure 1—figure supplement 2). There are only two arginine residues in the mDia2^NES^ peptide, Arg1189 and Arg1191. If the long side chain density is assigned to Arg1189, then Arg1191 would end up pointing into the binding groove – a very energetically unfavorable and unlikely situation. Furthermore, Ala1188 would end up in the P2 pocket of CRM1 where there is obvious density for a longer hydrophobic side chain (left panel of Figure 1—figure supplement 2). On the other hand, when Arg1191 is assigned to the long and continuous side chain density (adjacent to helix H12A of CRM1), the remaining side chains in the NES end up in positions that are consistent the electron densities.

For the FMRP-1b^NES^ (sequence: ^1^GGS-YLKEVDQLRALERLQID^20^), there are no obvious long side chain densities that could help with modeling. There are however obvious densities for several side chains in the first two turn of the NES helix. These side chain densities are consistent with two possible sequence assignments: ^5^LKEVDQLRAL^14^ or the more C-terminal ^11^LRALERLQID^20^. We tested modeling of FMRP-1b^NES^ by refining both peptide models and by testing a mutant peptide that should distinguish between the two models. Ten cycles of PHENIX refinement of the ^5^LKEVDQLRAL^14^ model resulted in positive and negative difference densities (mFo-DFc map) at several NES side chains, which suggested an incorrect assignment (left panels, Figure 1—figure supplement 2). In contrast, difference densities are absent when the ^11^LRALERLQID^20^ model is refined (right panels, Figure 1—figure supplement 2). The final FMRP-1b^NES^ structure was therefore modelled as ^11^LRALERLQID^20^.

The sequence assignment of FMRP-1b^NES^ was also tested in an additional experiement using a mutant FMRP-1b^NES^ that has the sequence YLKEVDQLRALER. If the NES is ^5^LKEVDQLRAL^14^, FMRP-1b^NES^ mutant YLKEVDQLRALER should bind well to CRM1. However, if ^11^LRALERLQID^20^ is the FMRP-1b^NES^, mutant YLKEVDQLRALER should not bind CRM1 as the C-terminal half of the NES or ^17^LQID^20^ which includes Ф3 and Ф4 is missing. New experimental results in Figure 1—figure supplement 3 show that FMRP-1b^NES^ mutant YLKEVDQLRALER does not bind CRM1, providing further support that the NES is indeed ^11^LRALERLQID^20^ as currently assigned.

We also agree with the reviewers that B-factors of the NES peptides, which are mostly between 60-90 Å2, with the expection of CDC7^NES^-ext and Hxk2^pep^ at 90-100 Å2, are high compared to the overall average B-factors for the proteins in the crystalsin all structures). We examined the average B-factors for 29 CRM1 residues that line the NES groove, and noticed that they are in the 50-70 Å2 range, higher than average B-factors for the rest of CRM1 but lower than NES B-factors. The high NES B-factors are likely due to partial occupancy of the NES peptides in the CRM1 grooves. Our peptide models remain reliable despite high B-factors as kick OMIT densities calculated with NES peptides omitted clearly show presence of the NES peptides and the omit densities also match the final peptide models well (Figure 1—figure supplement 4 and Figure 2—figure supplement 1).

Changes in the text of the manuscript:

We added a sentence in main text of the manuscript (Results section) to refer to the Methods section for details on our modeling strategy: “All eight NESs bind *^Hs^*CRM1 in the presence of RanGTP with dissociation constants (K_D_s) of 670 nM-20 μM, and were crystallized bound to the previously described engineered *^Sc^*CRM1-RanGppNHp-Yrb1p complex (Figure 1—figure supplement 1, [Supplementary-material SD1-data], [Supplementary-material SD2-data], [Supplementary-material SD3-data]) (Xu et al., 2012). Details of how the NES peptides were modelled can be found in the Methods section and in Figure 1—figure supplement 2, Figure 1—figure supplement 3.”

We also added the following description of how peptides are modeled to the Methods section:

“NES peptides are modeled according to the positive difference density (2mFo-DFc map) at the binding groove after refinement of the CRM1-Ran-RanBP1 model. […] Results in Figure 1—figure supplement 3 show that FMRP-1b^NES^ mutant YLKEVDQLRALER does not bind CRM1, providing further support that the NES is indeed ^11^LRALERLQID^20^ as currently assigned.”